# Hydrostatic Pressure Wheel for Regulation of Open Channel Networks and for the Energy Supply of Isolated Sites

**Ludovic Cassan** [1,*], **Guilhem Dellinger** [2], **Pascal Maussion** [3] and **Nicolas Dellinger** [2]

1   Institut de Mécanique des Fluides, Toulouse INP, Université de Toulouse, Allée du Professeur Camille Soula, 31400 Toulouse, France
2   Laboratoire des Sciences de l'Ingénieur, de l'Informatique et de l'Imagerie (ICUBE), ENGEES, 67000 Strasbourg, France; guilhem.dellinger@engees.unistra.fr (G.D.); dellinger@unistra.fr (N.D.)
3   LAPLACE, Toulouse INP, Université de Toulouse, 2 Rue Camichel, 31071 Toulouse, France; pascal.maussion@toulouse-inp.fr
*   Correspondence: lcassan@imft.fr

**Abstract:** The Hydrostatic Pressure Wheel is an innovative solution to regulate flow discharges and waters heights in open channel networks. Indeed, they can maintain a water depth while producing energy for supplying sensors and a regulation system. To prove the feasibility of this solution, a complete model of water depth–discharge–rotational speed relationship has been elaborated. The latter takes into account the different energy losses present in the turbine. Experimental measurements achieved in IMFT laboratory allowed to calibrate the coefficients of head losses relevant for a large range of operating conditions. Once the model had been validated, an extrapolation to a real case showed the possibility of maintaining upstream water level but also of being able to produce sufficient energy for supplying in energy isolated sites. The solution thus makes it possible to satisfy primary energy needs while respecting the principles of frugal innovation: simplicity, robustness, reduced environmental impact.

**Keywords:** renewable energy; water wheel; low tech; frugal innovation; experimental model; theoretical model

## 1. Introduction

Irrigation systems have a lot of weir structures in order to change the water elevation, water velocities, etc. These "small" drops represent a non-negligible hydroelectric potential [1]. Nevertheless, the role of the turbine in this study is a little bit different. Its function is not only to produce electricity but also to serve as a water level controller. Indeed, the regulation of water height and flow discharge in irrigation systems is also crucial to save the water resources and, as mentioned before, requires many regulation systems (sluice gates, spillways, weirs, etc). The fact is that these systems can be very isolated and therefore difficult to power, especially in remote areas. To overcome this problem, a turbine could regulate the water level automatically by adjusting its rotational speed. Part of the energy supplied by the turbine would then be used to power the sensors and the control system needed for this regulation and data transmission. Moreover, in the context of isolated areas, the rest of the energy provided by the turbine could also be used for supplying the local electricity grid. Indeed, the water wheels are also a good example of the low-tech concept for energy supply [2]. Therefore, we will also discuss the power performance in the context of isolated communities in developing countries in line with the frugal innovation concept. Although equipping a low head in a large river may involve some environmental issues, power generation with low-head coupled with solar energy and storage devices is fully relevant if no other source is available (isolated zone).

As noted above, irrigation systems are constructed with weirs that can be exploited. Typically, these structures have head differences between 0.5 and 3 m and have a low

flow rate. Turbines suitable for this kind of hydraulic sites are the Kaplan turbine, the Archimedean Screw Turbine (AST) and the various types of water wheels [3]. It is also possible to add the Very Low Head (VLH) turbine to this list. For obvious economical and ecological reasons, the turbine chosen must be inexpensive, robust, fish-friendly and provide the sediment continuity. The Kaplan and VLH turbine reach a better hydraulic efficiency than the other two, but they require complex control elements and are much more expensive [4]. The AST turbine has almost the same efficiency as a well-designed water wheel but is slightly more expensive and much more complex to build. Finally, the choice was made to consider water wheels.

By combining the different types of water wheels, they are able to exploit water head from 0.5 to 12 m [5]. The three main types of water wheels are the overshot wheels, the breastshot wheels and the undershot wheels. It is also possible to speak about stream water wheels but they are designed to exploit the flow velocity and not the head difference [6]. For the overshot wheels, the water enters from above and the wheel is rotated by the water weight. These turbines are used for heads between 2.5 and 12 m and can reach efficiencies of about 80% [4,7]. In the case of the breastshot wheels, the water enters at approximately the same height than the turbine's rotation axis. This type of wheel works for head differences between 1.5 and 4 m and can achieve efficiencies between 60% and 70% [8–11]. The last main type of water wheel is the undershot wheel. In this case, the flow enters below the axis of rotation. This type of turbine is adapted for very low head differences between 0.5 and 2.5 m and can reaches efficiencies of 80% [12,13]. Regarding the head differences in the context of irrigation systems, undershot wheels seem to be the most suitable. Specifically, the Hydrostatic Pressure Wheel (HPW) developed by [14] is chosen in order to achieve the objectives presented above. The wheel is composed of radial blades and is driven to rotate by the hydrostatic pressure exerted by the flow on the blades. In addition, this turbine is chosen because it presents a very simple design, is robust, inexpensive and fish-friendly [14].

As a reminder of the general context, the stated objectives are to replace the regulation structures of irrigation system that require manual operation and maintenance with a turbine. The latter will adjust the water level automatically by changing its rotational speed and, at the same time, will produce some energy for local consumption. In order to properly design the system according to the site characteristics, a theoretical model that links hydraulic conditions, geometrical parameters and turbine performance is required. Currently, there is only one theoretical model, proposed by [14]. However, this model is not able to predict directly the water level upstream to the wheel and the flow discharge through it. In irrigation systems, due to the modulation of water demand, it is essential to control water levels and flows throughout the network. Therefore, to control these parameters with water wheels, an improved model is needed and is presented in this article.

The paper is structured as follows. In Section 2, a new theoretical model is established based on the [14] model. Special attention is paid to the consideration of various energy losses (gap leakage, drag forces, etc.) as they are very significant. The experimental setup installed in the IMFT hydraulic laboratory and used for providing experimental data on the HPW turbine is presented in Section 3. After calibrating the loss models with the experimental results, the ability of the theoretical model to accurately reproduce the wheel performance, the upstream water levels and the flow discharges absorbing by the wheel are discussed in Section 4. Finally, a discussion about the possibilities to use these turbines for the electrification of an isolated village in a low-tech context is analyzed in Section 5.

## 2. Operating Principle and Theoretical Models

### 2.1. Operating Principle

As shown in Figure 1, the HPW developed in [14] consists of a wheel made up of radial blades longer than the inlet water height. This turbine converts the potential energy of a fluid into mechanical energy thanks to rotation induced by the fluid pressure exerted on its blades. This mechanical energy can then be transformed into electrical energy thanks

to a generator. This last transformation will not be discussed here although it is a crucial point for power generation [14]. The mechanical power supplied by the wheel is given by:

$$P = P_{hyd}\,\eta = \rho g Q \Delta H \eta \tag{1}$$

where $P_{hyd}$ is the available hydraulic power, $\rho$ is the density of water, $Q$ is the flow rate, $g$ is the acceleration of gravity, $\Delta H$ is the hydraulic head difference and $\eta$—the hydraulic efficiency of the turbine. The efficiency depends on the different losses present in the turbine which are mainly due to the flow leakages and the drag forces. The latter present when the blades enter into the flow and when they leave it. Thus, minimizing these losses is a key issue in order to increase the turbine performance.

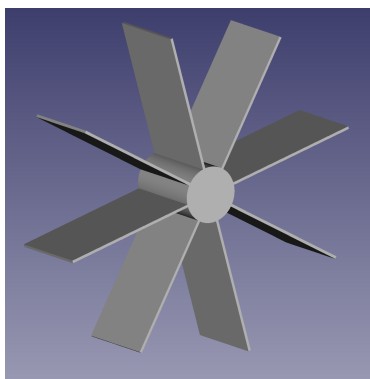

**Figure 1.** 3D view of a HPW turbine.

To use the wheel effectively as a regulation and energy production system, it is necessary to determine the influence of the wheel on the upstream water level and to predict its energy recovery performance. For this purpose, a theoretical model is established in the following part. The latter is able to determine the upstream water level and the performance of the turbine according to the geometrical parameters of the wheel and the flow conditions.

### 2.2. Theoretical Model

As presented in Figure 2, the HPW turbine has radial blades longer than the inlet water level. This is why this type of wheel is dedicated to very low heads between 0.3 and 1 m [14]. The main geometrical parameters of this turbine are the radius of the wheel $r$, the width $L$ and the number of blades $N$. These parameters are shown in Figure 2. It should be noted that each blade is defined by a plane that contains the axis of rotation. The mechanical power is obtained by the displacement of a vertical blade submitted to a force induced by the flow pressure more important on the upstream face than on the downstream side. The model is derived by considering force balance on a blade. Compared to the previous study [14], the theory is rewritten here with a global efficiency formula $\eta$ incorporating leakages and turbulence disturbance. The theory presented here will help upscale the wheel in a real scale application. Furthermore, as discussed earlier, an integrated formula for mechanical power as a function of upstream water depth and rotational speed is needed to develop efficient control of water level. To establish the model, a vertical blade with a finite radius $r$ and a width $L$ is considered. The water level upstream to the blade is equal to $d_1$ and the downstream one is equal to $d_2$. The distance between the blade and the bed is denoted $d_l$ (Figure 2).

The total flow rate $Q$ can be decomposed in two parts:

$$Q = Q_w + Q_l \tag{2}$$

with $Q_w$ being the flow through the surface bounded by the height $d_1$ and the width $L$ that actually contributes to wheel rotation, and $Q_l$ representing the flow leakage that appears

around the wheel. It may be noted that to optimize the wheel performance, it is necessary to limit the value of $Q_l$.

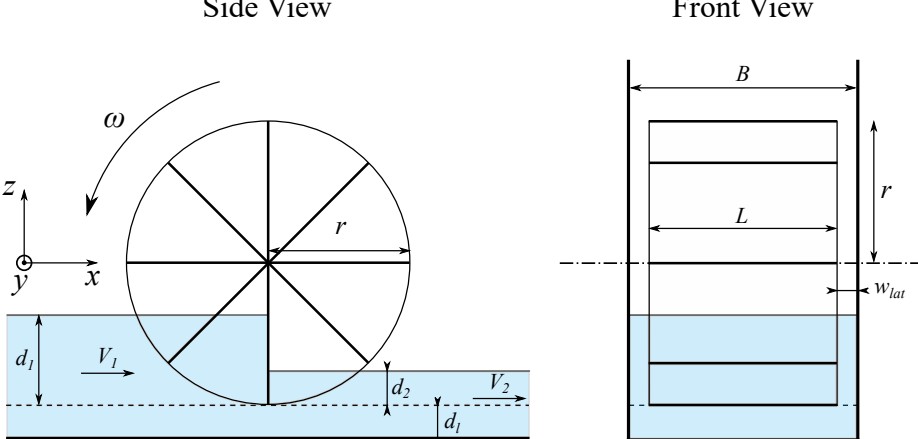

**Figure 2.** Schematic view of the HPW machine.

The flow leakage $Q_l$ can be decomposed in two leakages. The first one is $Q_{l,1}$, which occurs between the blades and the bottom and, the second one—$Q_{l,2}$, which occurs between the wheel and the two side walls. Both leakages can be determined by using the Torricelli equation and are a function of the head difference $(d_1 - d_2)$, the wheel width $L$ and the gaps between the wheel and the canal. To determine the first leakage $Q_{l,1}$ occurring between the blades and the bottom, the gap $d_l$ is not used directly. Instead, an equivalent opening $w$ representing the surface opening for flow under the wheel is introduced. This value has to be calibrated for each wheel geometry for the following reasons. Firstly, unlike sluice gate, the discharge formula is written here without considering the vertical contraction of the flow stream. Indeed, it is difficult to measure and evaluate accurately this value knowing that it evolves with the submergence $d_2/d_1$ [15,16]. Secondly, the wheel cannot be modeled like an orifice because the blades are moving and then the opening depends on the radial position of the wheel. Therefore $w$ represents an averaged value of the space between blade and bed during a rotation. Eventually, to determine the leakage $Q_{l,2}$, the gap $w_{lat}$ between the wheel an the sidewall is used. As a consequence, the portion of the discharge $Q_l$ that was not flowing into the wheel can by expressed as:

$$Q_l = \sqrt{2g(d_1 - d_2)}wL + 2\sqrt{2g(d_1 - d_2)}d_1 w_{lat} \tag{3}$$

The water flowing into the wheel $Q_w$ can be determined by assuming that when a blade is in the vertical position in the fluid, the water and the blade have the same velocity locally. The submerged length of the blade is equal to $d_1$, as represented in the Figure 2. The averaged velocity of the flow $v_a$ is then given by the integration of the horizontal component of the velocity of the submerged part of the blade:

$$v_a = \frac{1}{d_1} \int_r^{r-d1} z\omega dz = \omega r \left(1 - \frac{1}{2r/d_1}\right) \tag{4}$$

The flow $Q_w$ is then equal to:

$$Q_w = Ld_1 v_a = Ld_1 \omega r \left(1 - \frac{1}{2r/d_1}\right) \tag{5}$$

For scaling purposes, the following geometric aspect ratios are defined:

- water depths ratio: $X = d_2/d_1$;
- vertical gap ratio: $Y = d_l/d_1$;
- equivalent vertical gap ratio: $W = w/d_1$;
- horizontal gap ratio: $W_l = w_{lat}/B$;
- blade length ratio: $D = r/d_1$;
- wheel width ratio: $Z = L/B$.

The continuity equation provides the following relationship:

$$
\begin{aligned}
Q &= Q_w + Q_l \\
&= v_a d_1 L + \sqrt{2g(d_1 - d_2)}wL + 2\sqrt{2g(d_1 - d_2)}d_1 w_{lat} \\
&= L d_1 \omega r \left(1 - \frac{1}{2r/d_1}\right) + \sqrt{gd_1}\sqrt{2(1-X)}\,(wL + 2d_1 w_{lat}) \quad (6)
\end{aligned}
$$

The ratio $R_Q = Q_w/Q$ between the water flowing through the wheel and the total discharge is defined and equal to:

$$
R_Q = \left(1 - \frac{[WZ + W_l]\sqrt{2(1-X)}}{F_r}\right)\frac{1}{1 - \frac{1}{2D}} \quad (7)
$$

where $F_r = Q/(Bd_1\sqrt{gd_1})$ is the Froude number calculated in the upstream part of the canal. $R_Q$ is then the ratio of leakage ($R_Q = 1$ for no leakage).

The mechanical power recovered by the wheel is determined by multiplying the torque presents on the wheel axis by the rotational speed of the turbine. The torque can be decomposed in two parts: the drive torque induced by the pressure exerted by the water on the blades and, the resisting torque induced by the drag forces. The latter are mainly present when a blade enters or leaves the flow. To simplify the model, all drag forces acting on the wheel are modeled with a single overall drag force $F_{drag}$ that would be applied to the blade tip with a direction orthogonal to the blade surface. The drag forces were assumed to depend primarily on the rotational speed of the wheel, the flow velocity, both inlet and outlet water levels and turbine width. The question remained as to what velocity and area should be used to effectively model all of the drag forces acting on the blades. After several trials, it appears that the best way to model an overall drag force $F_{drag}$ is to use the blade tip velocity with the area given by the difference in water depth times the width of the wheel. The drag force $F_{drag}$ is then equal to:

$$
F_{drag} = \frac{1}{2}\rho(d_1 - d_2)L(r\omega)^2 C_d \quad (8)
$$

with $C_d$ a drag coefficient. The resistant torque induced by the drag force is then given by drag force $F_{drag}$ multiplied by the wheel radius $r$. The power $P_d$ is then given by the resisting torque multiplied by the rotation speed of the wheel $\omega$:

$$
P_d = F_{drag}.r\omega = \frac{1}{2}\rho C_d(r\omega)^3 L(d_1 - d_2) = \frac{1}{2}\rho gr\omega L d_1(1 - X)C_d F_\omega^2 \quad (9)
$$

where $F_\omega = r\omega/\sqrt{gd_1}$.

To analyze the wheel performance, it is necessary to give the expression of the hydraulic efficiency $\eta$. As it can be seen in Equation (1), this efficiency corresponds to the ratio between the mechanical power recovered by the wheel $P$ and the available hydraulic power $P_{hyd}$. To determine the hydraulic head $\Delta H$ that appears in the expression of $P_{hyd}$, it is assumed that the kinetic energy is computed from a zone of uniform flow in the upstream and downstream part of the canal. This means that the energy losses at the inlet and at the outlet of the wheel are taken into account in the efficiency. Considering a flat bottom near the wheel, the expression of $\Delta H$ is then equal to:

$$\Delta H = d_1 - d_2 + \frac{Q^2}{2gB^2}\left(\frac{1}{(d_1 + d_l)^2} - \frac{1}{(d_2 + d_l)^2}\right) \tag{10}$$

The mechanical power $P$ provided by the wheel corresponds to the available hydraulic power calculated with the flow $Q_w$ minus the drag loss $P_d$:

$$P = \rho g Q_w \Delta H - P_d = \frac{1}{2}\rho g Q d_1\left((1 + X) - \frac{1}{3D}\left(1 + X^2 + X\right) - C_d F_\omega^2\right)(1 - X)R_Q \tag{11}$$

Finally, combining the Equations (7), (9) and (10); the wheel efficiency $\eta$ is equal to:

$$\eta = \frac{1}{2}\frac{(1 + X) - \frac{1}{3D}\left(1 + X^2 + X\right) - C_d F_\omega^2}{1 - \frac{1}{2}F_r^2\frac{1 + X + 2Y}{(1 + Y)^2(X + Y)^2}}R_Q \tag{12}$$

Except the drag term, the Equation (12) is similar to the formula proposed by [14]. Here all corrections are given in the same equation which provides a better view of the dimensionless significant numbers. These independent numbers are $X, Y, Z, C_d, D, F_r, \omega$. The number $R_Q$, $F_\omega$ and $\eta$ can be deduced from others. For a given rotational speed $\omega$, the Froude number provides the relationship between flow rate and water depth. The coefficient $C_d$ is considered constant for a given wheel shape even though a Reynolds dependence could be expected. This analysis tends to prove that a Froude similarity could be applied for the wheel design.

## 3. Experimental Setup

Experimental data are necessary to calibrate and validate the theoretical model established previously. In addition, experimental measurements are needed to investigate the performance of the HPW turbine experimentally. An experimental setup was therefore installed in the hydraulic laboratory of the Toulouse Institute of Fluid Mechanics (IMFT). This device allows to test the efficiency and the mechanical behavior of the turbine for different geometrical and hydraulic parameters. The wheel geometries considered in this study are as simple as possible in order to obtain low-cost and low-maintenance turbines. These characteristics are essential in a context of isolated sites where it is difficult to replace a broken component with a new one. Moreover, plane blades seem to be more suitable for controlling the flow through the wheel even at low speeds by stopping it completely if necessary. The blades are attached to the side disks. They are slightly shorter than the radius which allows the air and water pressures to be balanced between the blades.

Eventually, the geometrical parameters of the tested wheel and the flow conditions are shown in Table 1.

**Table 1.** Geometrical and hydraulic parameters of the experimental device.

| Geometrical Parameters | Outer radius—$r$ (m) | 0.4 |
|---|---|---|
| | Material | PVC |
| | Width—$L$ (m) | 0.2 |
| | Number of blades | 8 |
| | Channel width—$B$ (m) | 0.4 |
| | Vertical gap—$d_l$ (m) | 0.025 and 0.05 |
| | Horizontal gap—$w_{lat}$ (m) | 0.002 |
| Flow conditions | Flow rate—$Q$ (m$^3$ s$^{-1}$) | 0.005 ... 0.025 |

The wheel is installed in a 4 m long, 0.4 m wide and 0.4 m high open channel. Aluminum walls of 0.9 m width are placed on each side of the wheel to conduct the flow into the turbine. The bed and the wall of the channel are in glass. The slope of the bed is chosen to be zero. Water is brought at the inlet of the experimental device by a centrifugal pump. The flow rate is measured with a electromagnetic flowmeter (IFS 4000/6, Krohne, Holland) with a given measuring accuracy of ±0.5%. The water then flows through the wheel and

leads to its rotation. The water then exits at the outlet. All water levels are measured with dial gauges. The torque provided by the wheel $C_h$ is balanced by an overall brake torque, which consists of two terms: a friction term $C_f$ induced by the friction in the device (rotational guidance) and a brake torque $C_b$ induced and controlled by a Prony brake. The motion equation is defined as follows:

$$J\frac{d\omega}{dt} = C_h - C_f - C_b \tag{13}$$

with $J$—the inertia of the rotating parts. A torque meter is coupled directly in line, between the wheel axis (accuracy of 1 mN m) and the Prony brake. This latter provides the value of $C_b$; in steady state the equation above becomes:

$$C_b = C_h - C_f \tag{14}$$

Note that the instantaneous values of torque and rotational speed are averaged over 30 s, a sufficient value considering the frequency of the dynamic phenomenon. $C_f$ was determined for different values of $\omega$ without water in the channel. It appears that this torque increases linearly with the speed and its value remains close to zero when $\omega = 0$. The rotational speed of the turbine is measured with the incremental coder incorporated in the torque meter (T21WN, HBM, Germany). Finally, the mechanical power provided by the wheel is determined by multiplying the torque by its rotational speed. A schematic diagram of the whole device is shown in Figure 3.

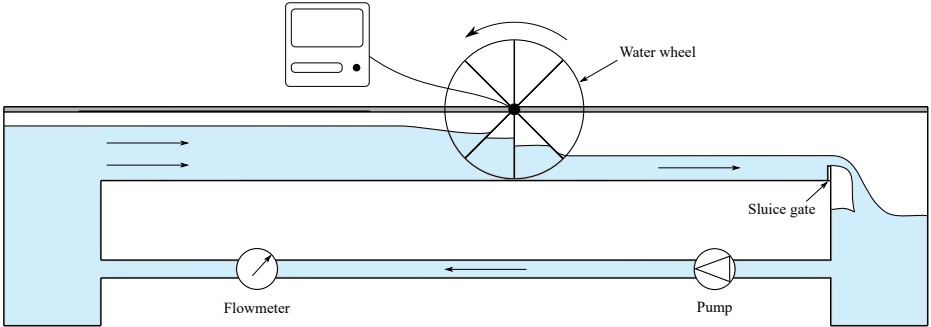

**Figure 3.** Schema of the experimental device with the HPW wheel without a removable bottom.

In the experiments, two configurations were tested. In the first, called $C_1$, there is a vertical gap $d_l$ of 0.025 m between the blade and the bed of the channel. This configuration was chosen to determine the value of the equivalent gap $w$ in the flow leakage equation. In the second configuration, named $C_2$, the wheel axis is placed at a higher position. The gap $d_l$ is then equal to 0.05 m. Nevertheless, a block of aluminum (0.4 m $\times$ 0.045 m $\times$ 0.9 m) is placed just below the wheel, as shown in Figure 4. This block can be considered as a simplified shape of the curved shroud proposed in [14]. In this case, the gap between the block and the blade of the wheel is equal to 0.01 m. This important gap leads to a strong flow leakage. However, the objective of this study is not to obtain the best hydraulic efficiencies but to be able to correctly quantify this leakage and its influence on the performance. Moreover, this configuration represents the case of nonideal implementation that can be found in isolated areas. Lastly, the wheel axis being positioned higher in the $C_2$ configuration than in the $C_1$ one, it appears that the downstream water level was under the wheel for some operating points. A photo of the experimental device with the HPW turbine can be seen in Figure 5.

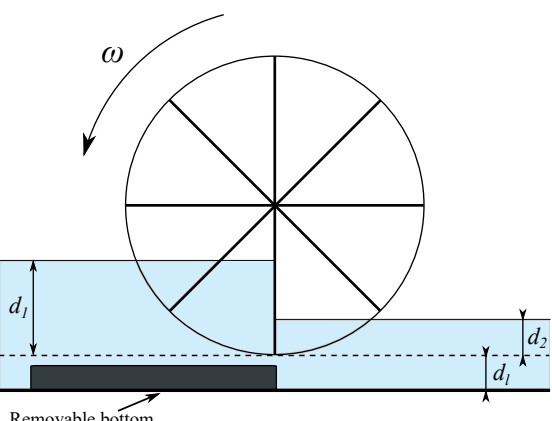

**Figure 4.** Schematic view of the HPW with the removable bottom in the configuration $C_2$.

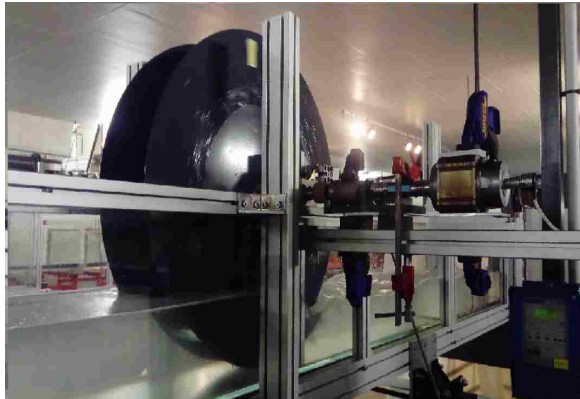

**Figure 5.** Photo of the experimental device with the HPW wheel (configuration ($C_1$)).

## 4. Results

### 4.1. Model Calibration

Flow leakage is the main energy loss in HPW wheels. In order to accurately determine the wheel performance, it is necessary for the theoretical model to best reproduce this loss. As can be seen in Equation (3), the flow leakage $Q_l$ is a function of an equivalent opening $w$. This opening must then be calibrated with experimental data. To do so, the experimental and theoretical values of $R_Q$ are compared for several operating points, i.e., for various hydraulic (flow rate, water levels, etc.) and geometrical parameters (configurations $C_1$ and $C_2$). The theoretical values of the ratio $R_Q$ are calculated from Equation (7). The latter involves the constant $W = w/d_1$ and the Froude number $F_r$. The first one must be calibrated and the second one is measured experimentally. The other constants in Equation (7) are known. The experimental values of $R_Q = Q_w/Q$ are determined by calculating $Q_w$ from Equation (5) and by measuring the flow rate $Q$. Indeed, as demonstrated above, the flow $Q_w$ is a function of known geometrical parameters and of the rotational speed, which is measured experimentally.

Figure 6 exposes the comparison between modeled and measured values of $R_Q$. It can be seen that the theoretical model reproduces the measured data well. The model is more accurate in configuration $C_1$ for which the maximum deviation is 10%. This can the explained by the fact that the block of aluminum placed behind the wheel in configuration $C_2$ perturbs the flow pattern, especially for free-flow experiments ($X = 0$). The calibrated values of $w$, the lateral gap $w_{lat}$ and the vertical gap $d_l$ are shown in Table 2 for the $C_1$ and $C_2$ configurations. For the $C_1$ configuration, the value of $w$ is greater than the gap $d_l$ between the channel bottom and a vertical blade. This can be explained by the fact that the gap increases when the considered blade rotates and is not vertical anymore. For the $C_2$ configuration, the value of $w$ is lower than $d_l$ because of the presence of the block under the

wheel. Interestingly, although the $d_l$ deviation in the $C_1$ and $C_2$ configurations is different, the calibrated values of $w$ are both very similar. This could be due to the fact that the wheel has only eight blades. Most of the leakage then flows between the blades. It is expected that the value of $d_l$ should have more influence on the wheel performance if it has more blades.

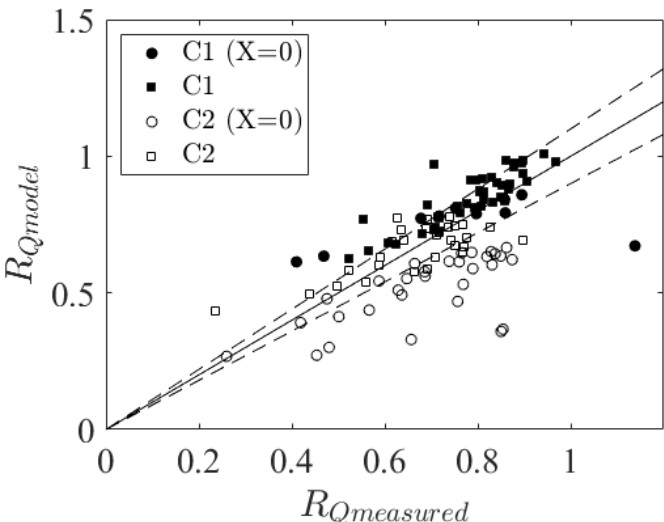

**Figure 6.** Comparison between modeled and experimental values of $R_Q = Q_w/Q$ for configuration $C_1$ (without removable bottom) and $C_2$ (with removable bottom). The solid line represents the perfect agreement and the dashed lines represent the gap of 10%.

**Table 2.** Measured values of the lateral $w_{lat}$ and vertical gap $d_l$, calibrated value of the equivalent opening $w$ and of drag coefficient $C_d$ for configurations $C_1$ and $C_2$.

|       | $d_l$ (m) | $w_{lat}$ (m) | $w$ (m) | $C_d$ |
| ----- | --------- | ------------- | ------- | ----- |
| $C_1$ | 0.02      | 0.002         | 0.026   | 4     |
| $C_2$ | 0.05      | 0.002         | 0.025   | 4–10  |

The power loss $P_d$ induced by the drag force and the associated turbulent dissipation is given by the Equation (9). In this equation, the drag coefficient $C_d$ must be calibrated with experimental data. To do this, special attention is paid to the drag term $C_d F_\omega^2$ presents in Equation (11). The experimental value of this drag term is obtained by measuring experimentally the other terms ($P$, $Q$, $\omega$, $d_1$ and $d_2$) present in Equation (11) and by using the values of $w$ calibrated previously (Table 2). Figure 7 exposes the evolution of the drag term as a function of $F_\omega$ for the experimental configurations $C_1$ and $C_2$. It can be shown that the dissipation depends on the square of $F_\omega$ and can be expressed as the proposed modeling $C_d F_\omega^2$. As shown in Table 2, it was possible to deduce from experiments a constant $C_d = 4$ for configurations $C_1$ and $C_2$ when $F_\omega$ is greater than 0.3. For configuration $C_2$, the agreement is worse at low $F_\omega$ and the value of $C_d$ is higher ($C_d = 10$). This is explained by the flow disturbance induced by the block placed under the wheel. It can be noted that for very low values of downstream water level (i.e., for $X = 0$), the flow is almost similar to a free flow over a weir. It differs from the assumption made to establish the theoretical analysis and an additional dissipation appears then. Although this behavior is partially taken into account by the term $(d_1 - d_2)L$ representing the drag area, this correction is not sufficient. To better understand the flow pattern, 3D numerical simulations are planned for the future.

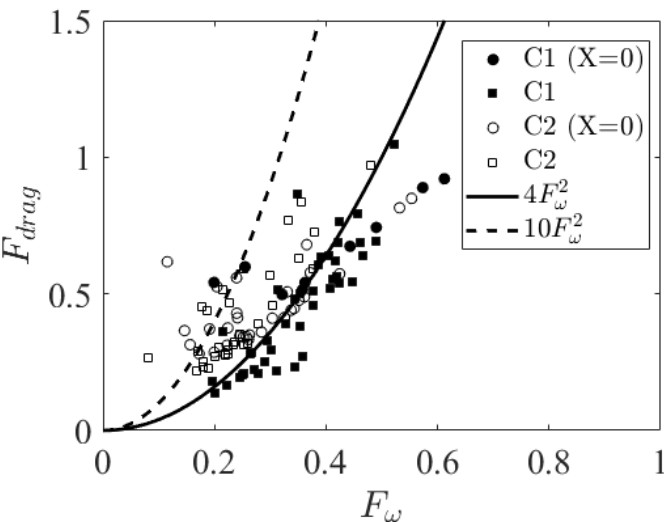

**Figure 7.** Evolution of the drag term $C_d F_\omega^2$ depending on $F_w$ for configurations $C_1$ and $C_2$.

### 4.2. Power and Efficiency

Figures 8 and 9 show the comparison between experimental and modeled values of the inlet water level $d_1$ for the $C_1$ and $C_2$ configurations, respectively. The theoretical values of $d_1$ are determined using the calibrated values of $w$ and $C_d$ (Table 2). As expected, it can be shown that the value of $d_1$ decreases when the rotational speed increases because the flow capacity of the wheel increases with the rotational speed. Figure 8 shows that the model can reproduce the experimental values in configuration $C_1$ accurately. For the $C_2$ configuration, the theoretical values of $d_1$ are slightly higher than experimental values (Figure 9). Again, this difference can be explained by the flow disturbance induced by the block. Nevertheless, these results show that the model is capable of predicting the water level upstream of the wheel. This ability is very important for the control of water levels and flows in irrigation networks.

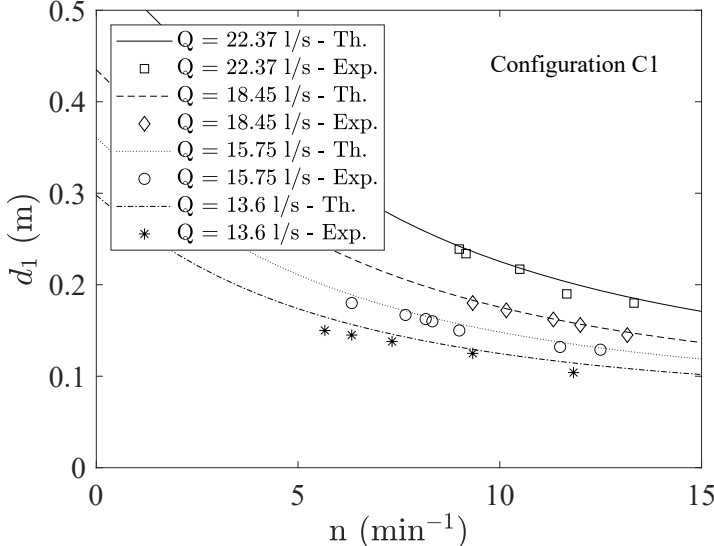

**Figure 8.** Comparison between modeled and experimental values of the inlet water level $d_1$ for the configurations $C_1$.

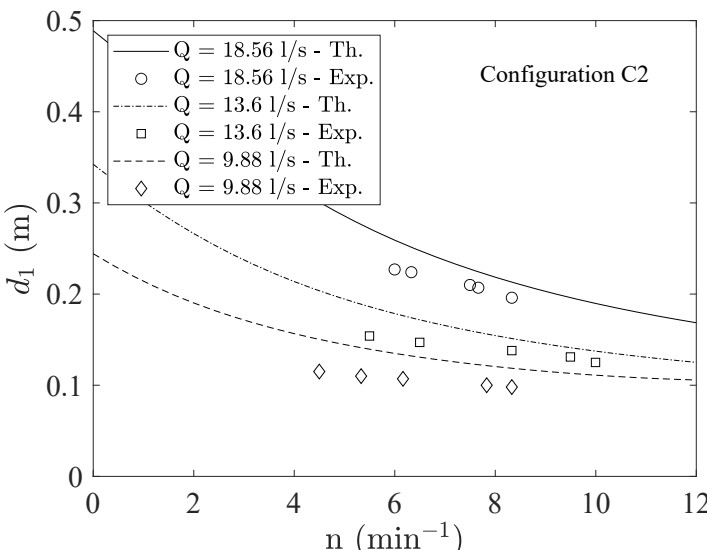

**Figure 9.** Comparison between modeled and experimental values of the inlet water level $d_1$ for the configurations $C_2$.

Figures 10 and 11 show the evolution of the mechanical power supplied by the wheel as a function of the rotational speed for the $C_1$ and $C_2$ configurations, respectively. The theoretical power evolution is determined by the Equation (11). It should be noted that for a given value of $d_1$ and $Q$, the rotational speed can be directly deduced from Equation (7). Then at a fixed rotational speed, a single value of $d_1$ corresponds to one discharge if $d_2$ is constant. It can be seen that the evolution of power is bell-shaped. Thus, for low rotational speed, the principal loss comes from flow leakages. Conversely, for too-high rotational speeds, the loss induced by turbulence and friction become predominant. As expected, there is an optimal rotational speed of the wheel that provides the best wheel performance. Comparison between theoretical and experimental values exposes that the model is able to determine the power delivered by the turbine, especially for the $C_1$ configuration (blockless configuration).

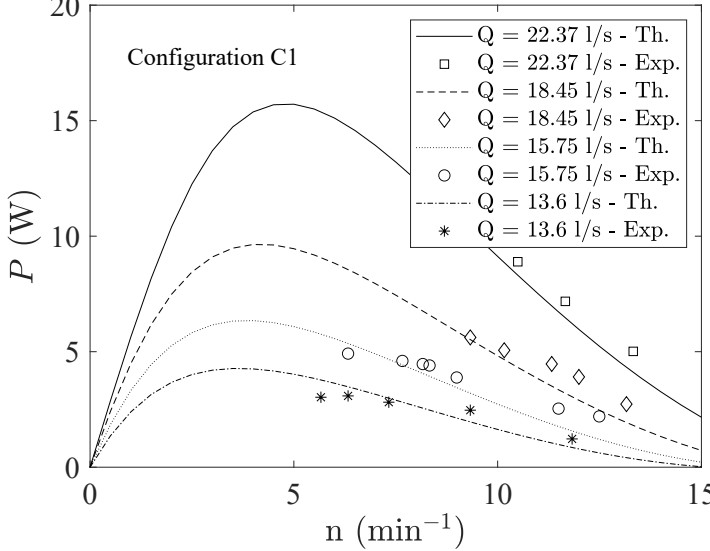

**Figure 10.** Theoretical and experimental values of mechanical power depending on the rotational speed for different flow discharges $Q$ and for configuration $C_1$.

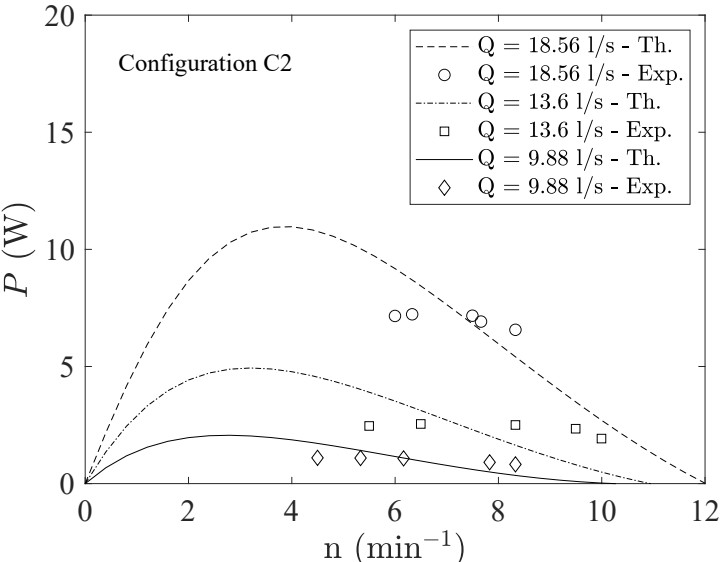

**Figure 11.** Theoretical and experimental values of mechanical power depending on the rotational speed for different flow discharges $Q$ and for configuration $C_2$.

Figures 12 and 13 expose the evolution of the hydraulic efficiency $\eta$ as a function of the rotation speed $n = 60/(2\pi)\omega$ for different flow rates. The theoretical efficiency is calculated from Equation (12). The experimental values are obtained by dividing the measured mechanical power by the hydraulic power according to the flow rate, $d_1$ and $d_2$. As for the mechanical power, a bell-shaped evolution of the efficiency can be shown. Comparing the modeled and experimental efficiencies for the $C_1$ configuration, it can be noticed that the model gives slightly higher efficiency values. The optimal rotational speeds of the wheel (i.e., the speed that gives the higher efficiency) given by the model are also close to the experimental ones for the $C_1$ configuration. For the $C_2$ configuration, the modeled efficiency values are much higher than experimental values because of the block under the wheel. When $n$ tends to 0, the efficiency also tends to 0 because the leakage flow rate becomes equal to the total flow rate ($R_Q \approx 0$). Indeed, the wheel behaves like a sluice gate and no mechanical power is obtained.

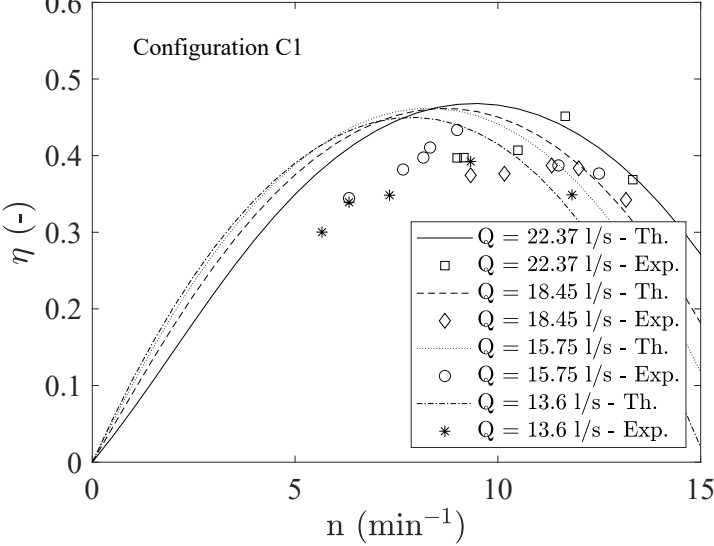

**Figure 12.** Theoretical and experimental values of hydraulic efficiency depending on the rotational speed for different flow discharges $Q$ and for the configuration $C_1$.

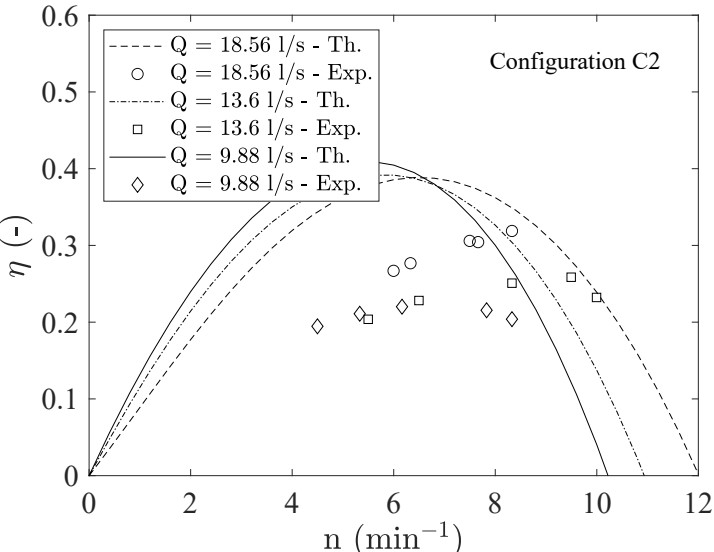

**Figure 13.** Theoretical and experimental values of hydraulic efficiency depending on the rotational speed for different flow discharges $Q$ and for the configuration $C_2$.

Regarding the experimental efficiency, it can be seen that the values are much lower than those presented in [14]. Indeed, while efficiencies higher than 80% are measured in [14], maximal values in this study are approximately of 45% (in the $C_1$ configuration), possibly because of the low number of blades—8 blades here and 12 in [14]. The loss by flow leakage is therefore much more important here. It can also be noticed that the $C_2$ configuration does not improve the HPW performance. The rectangular shape of the block appears to be too "simple" compared to curved blocks. Indeed, the leakage reduction is not significant with a rectangular shape of the removable bottom and the block-induced flow disturbances that dissipate energy.

## 5. Discussion

In this section, the previously established theoretical model is used to evaluate the ability of the hydraulic pressure wheel to achieve the two initial goals, i.e., to control the water level and to produce electricity in irrigation systems. To ensure the validity of the model, both low and high rotational speeds of the wheel are avoided. Indeed, very low velocities induce significant leakage that could be misrepresented by the model. Conversely, for too high rotational speeds, energy losses due to turbulence would not be modeled correctly. For the rest of the study, the number of blades and the distance between the blade and the bottom $d_l$ are respectively taken as equal to 8 and 0.025 m. These are the same values as in the experimental setup presented earlier. To approximate the equivalent opening $w$, it will be assumed that $w$ increases linearly with the wheel diameter.

### 5.1. Regulation of Water Levels

Firstly, the use of HPW in irrigation networks is considered in conjunction with the installation of a weir, as shown in Figure 14. It is then possible to limit the size of the device while ensuring sufficient spillage in case of too-important flows. The main advantage of using this turbine in an irrigation context is the fact that it is possible to supply energy locally. This can be used to control water levels (upstream and downstream), flow rates, etc. To do this, it is planned that the wheel supplies in electricity sensors and a regulation system that will adapt the speed of the turbine according to the desired values. In order to have a live coverage of the network, this energy can also be used for supplying wireless data transmission in addition to solar energy and storage devices. Other interests can be found in the use of HPW machines. For example, by ensuring an adequate design of the wheel, with tip blades close to the bed bottom, it is possible to ensure free flow of

sediments while avoiding silting. For anthropized natural systems, the water wheel can also ensure the free circulation of natural species present in the hydraulic system [17,18]. Eventually, HPW can also avoid the problem of floats usually present in the recirculation zone upstream of the gates.

To analyze the operation of the HPW system with respect to the water control objective, the example of a medium-sized network is chosen. The dimensionless Equations (7) and (12) ensure that the wheel behavior can be applied to a full-scale configuration. In the following, the case of a wheel of 2 m in diameter and 1 m in width is then considered. The dimensions of the device are shown in Table 3.

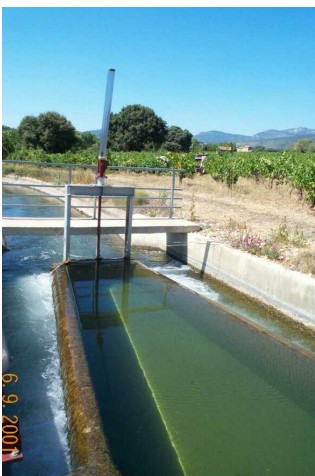

**Figure 14.** Example of sluice gate with a weir in an irrigation system.

**Table 3.** Geometrical parameters of the Hydraulic Pressure Wheel used for the discussion.

| Geometrical Parameters | Outer radius—$r$ (m) | 1 |
|---|---|---|
| | Width—$L$ (m) | 1 |
| | Number of blades | 8 |
| | Vertical gap—$d_l$ (m) | 0.025 |
| | Horizontal gap—$w_{lat}$ (m) | 0.02 |
| | Equivalent opening—$w$ (m) | 0.075 |

Figure 15 exposes the inlet water level $d_1$ as a function of flow rate $Q$ for different values of rotational speed $\omega$ and outlet water level $d_2$. It can be observed that a HPW machine of 2 m in diameter can regulate $d_1$ by changing $\omega$ up to about 0.8 m$^3$/s. It can also be noted that, for a given flow rate, the value of $d_1$ is very sensitive to the rotational speed. A wide range of flow rates (between 0.2 and 0.8 m$^3$/s) is covered for rotational speeds varying only by a factor of 3. This method is advantageous for coupling devices with already existing alternators. The influence of the downstream water level $d_2$ on the flow rate $Q$ is almost zero. This can be explained by the fact that the slight deviation comes from the flow leakage.

Figure 16 shows the mechanical power delivered by the wheel $P$ as a function of flow rate $Q$ for different values of rotational speed $\omega$ and outlet water level $d_2$. As expected, it can be seen that the power increases with the flow rate $Q$. Moreover, the power increases as $d_2$ decreases. This is because the head $H$ and thus the available hydraulic power $P_{hyd}$ increases as $d_2$ decreases for a given value of $d_1$. For the entire range of flow and speed, a few hundreds of watts can be produced. This production is sufficient to power a control and transmission system. It should be remember that, for this application, achieving maximum efficiency is not the main objective. However, as long as the ration $X = d_1/d_2$ is greater than 0.5, the efficiency remains greater than 30%. This value ensures a sufficient electrical production since the remaining power is also of the order of several hundred watts.

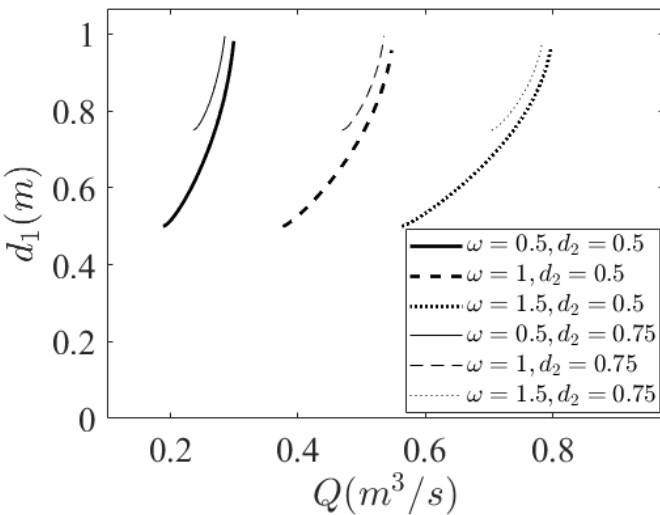

**Figure 15.** Simulated inlet water level estimated depending on the flow rate for different values of rotational speed and outlet water level.

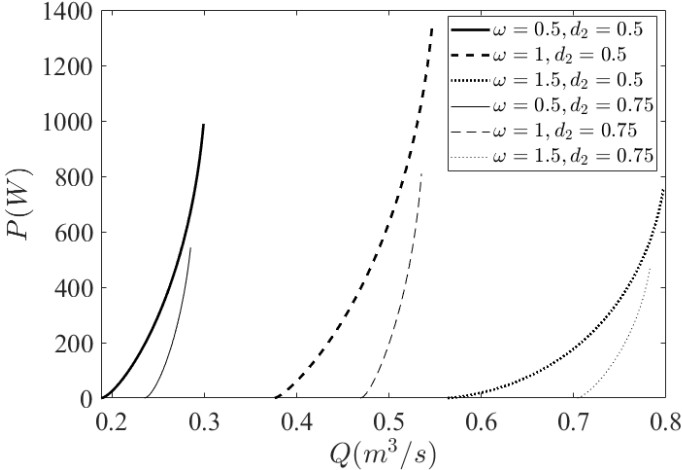

**Figure 16.** Simulated mechanical power estimates of the real-scale wheel depending on the flow rate for different values of rotational speed and outlet water level.

*5.2. Electric Production for Isolated Site*

Initially, the HPW wheel presented here is designed to produce electricity for low heads and low flows. For various environmental reasons, it seems judicious to limit this application to particular cases which benefit the most from the advantages of this system. These are its simplicity and robustness. Thus, run-of-river installations should be avoided to limit impacts and management of flood events. On the other hand, the diversion of high flows requires significant civil engineering costs. The interest is then focused on cases with low flows diverted (a few m$^3$/s) or small river courses where the wheel can be removed during floods. This context corresponds well to the electrification of isolated villages where even a few kilowatts can bring an important development aid [19].

As before, it is assumed that the HPW is installed in a 1 m channel (i.e., $L = 1$ m). It is then possible to work by unit width as one might do with a hill chart for a turbine. The other geometrical characteristics of the wheel are the same as those presented in Table 3. The outlet water level is taken equal to $d_2 = 0.7$ m.

In Figures 17 and 18, the operating diagram of the HPW is drawn in the $P, Q$ plans. The value of the inlet water level is limited by the wheel size ($d_1 < r$) and the rotational speed by $F_\omega = r\omega / \sqrt{g_d 1} < 1$. The evolution of the rotational speed depending on the flow discharge is nearly linear. The flow leakage has an influence only for low power and

then for low flow discharges. This is explained by low value of $d_l$ chosen in this example and by the fact that the ratio $Q_l/Q$ decreases as the rotational speed of the wheel increases. Despite the low efficiency values, which are generally lower than 50%, the results show that a wheel with a radius of one meter can provide 600 to 800 W per unit width. Again, these low efficiency values could come from the fact that the wheel has only 8 blades. Increasing the number of blades will increase the performance of the wheel (experimental data in [14]). It can be noted that the same powers can be obtained with the same upstream level. This is due to the fact that the efficiencies are higher for low rotational speeds. Indeed, for high rotational speeds, a lot of energy is dissipated by turbulence. Eventually, the maximum mechanical power is reached for $d_1 = r$, i.e., when the value of $X = d_2/d_1$ is minimal.

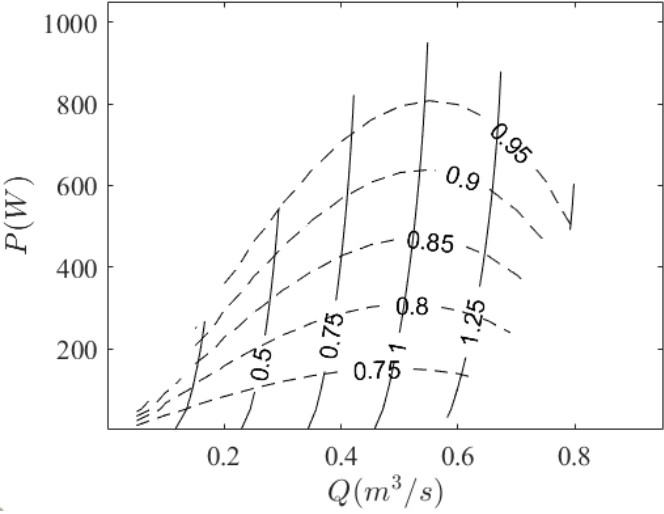

**Figure 17.** Iso-contours of rotational speed $\omega$ (solid line) and of upstream water level $d_1$ (dashed line) for the HPW with $d_2$ = 0.7 m.

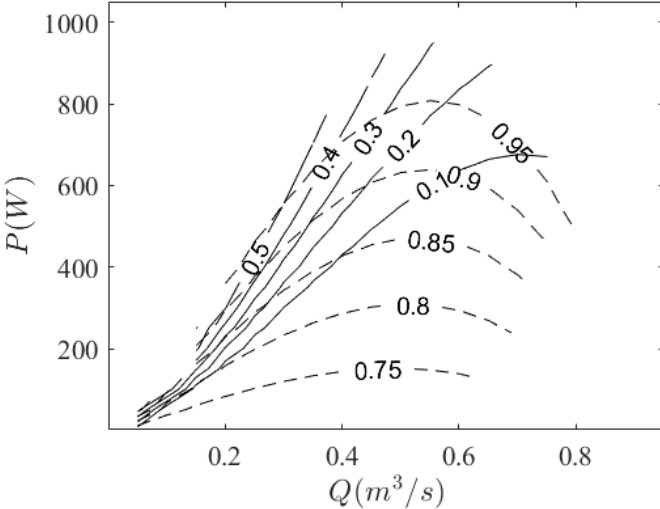

**Figure 18.** Iso-contours of efficiency $\eta$ (solid line) and of upstream water level $d_1$ (dashed line) for the HPW with $d_2$ = 0.7 m.

We recall that the results presented above are quite similar to those presented in [14] but with a different definition and calibration of dissipation terms. The contribution of this study concerns the quantification of flow losses, which is useful to properly design a "low cost" micro-power plant. Indeed, in a context of isolated areas where wheels can be manufactured directly on site, the bank heights are not always adapted to an optimal installation (i.e., $d_l = 0$). Moreover, the presence of large space under the wheel is an

advantage to limit the environmental impacts of the wheel (sediment transport and fish displacement). In the case of the previous wheel and for $h_u = 0.9$ m, $h_d = 0.8$ m, the influence of $d_l = w$ is studied. It appears that the recoverable power drops quickly when $d_l$ increases (Figure 19). For low flows, a few centimeters are enough to cancel the interest of the wheel. Nevertheless, for large flows (between 0.6 and 0.8 m$^3$/s), a consistent production of electricity can be maintained with values of $d_l$ close to 0.5 $r$. The minimum value of $d_l$ for a sedimentary and fish passage is of the order of 0.3 to 0.4 m [20]. These values provide sufficient production for the highest flows.

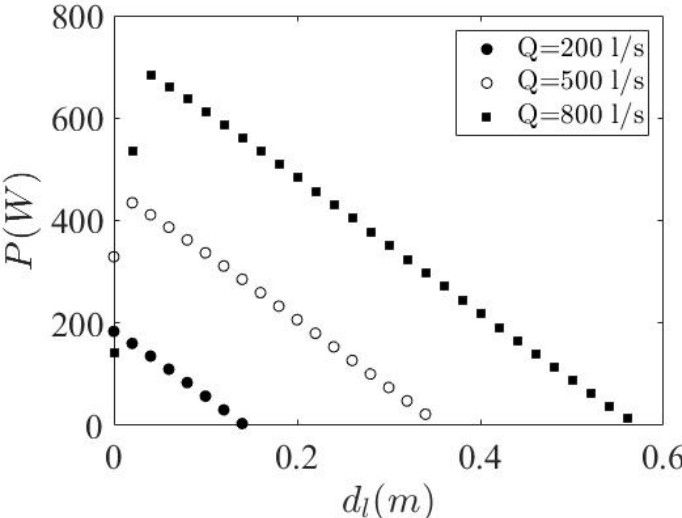

**Figure 19.** Influence of the gap between the wheel and the bottom $d_l$ on the mechanical power provided by the wheel.

## 6. Conclusions

Irrigation systems feature many hydraulic structures to change water elevations or flow velocities. These must be controlled according to the hydraulic conditions and the demand. This study was then focused on the use of a turbine to control these physical parameters by modifying its rotational speed. Moreover, this turbine could also be used to exploit the hydraulic energy present in this irrigation network. With controlled electricity storage, this energy production could be used to power sensors and provide electricity in the cases of isolated sites.

To achieve these objectives, the choice was made to use the Hydrostatic Pressure Wheel (HPW) developed by the authors of [21]. Indeed, although this technology does not have the best hydraulic performance, the HPW is simple, robust and low-cost. In order to properly design the wheel, a theoretical model relating the performance of the turbine to the hydraulic conditions is necessary. Therefore, this paper presents an improved theoretical model based on the one presented in [14]. An important novelty resides in the ability of the model to predict the inlet water level. This is particularly important in an irrigation network control context. Moreover, all energy loss terms are taken into account and modeled.

The model was then calibrated and evaluated with experimental data. The latter were provided by an experimental device installed in the IMFT laboratory. The comparison between modeled and measured values have shown that the model is able to reproduce the experimental results in a large range of hydraulic conditions. However discrepancies appear when the operating conditions are far from the hydrostatic functioning due to insufficient downstream water depth.

The extrapolation of the model to a real case has given indications on the installation of this type of turbine in irrigation systems. First, it was shown that a wheel with dimensions compatible with usual irrigation networks can produce between 100 and 1000 W per unit meter width. This power range is sufficient to power a remote control system. Moreover, an HPW's wheels can regulate the water level over a wide range of flow rates by adjusting

its rotation speed. Eventually, electricity production with a HPW can be significant even if there are high flow leakages due to implementation difficulties. Nevertheless, the mechanical power can drop quickly when the gap between the wheel and the bottom is too large. The turbine then no longer works with hydrostatic forces but recovers the kinetic energy.

**Author Contributions:** Conceptualization: L.C.; experimentations: L.C.; analysis: L.C., G.D., P.M. and N.D.; original draft preparation: L.C. and G.D.; review and editing: L.C., G.D., P.M. and N.D.; supervision: L.C. All authors have read and agreed to the published version of the manuscript.

**Funding:** This research received no external funding.

**Informed Consent Statement:** Not applicable.

**Data Availability Statement:** Not applicable.

**Conflicts of Interest:** The authors declare no conflict of interest.

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
