# Peer review of "Hydrostatic Pressure Wheel for Regulation of Open Channel Networks and for the Energy Supply of Isolated Sites"

_sustainability, doi:10.3390/su13179532_

Round 1

Reviewer 1 Report

The manuscript describes an analysis of the Hydrostatic Pressure Wheel as a hydropower converter for very low head differences. The HPW is analysed theoretically, including leakage and turbulent losses. Laboratory experiments of an HPW are described and the results analysed. The model had 8 blades, and no curved section at the bed so that leakage losses can be expected to be high. The results are then used to assess the hydropower potential in an irrigation canal.

  1. General comments:

The paper is well written and illustrated. There are some minor linguistic errors, so the authors are advised to check the manuscript with a native English speaker.

There are some corrections in the annotated manuscript.

The paper’s contribution is significant, giving a much more thorough theoretical background than provided in the previous literature and addressing several design relevant aspects such as drag losses and leakage which were not considered before. Also, simplified geometries which will probably occur in real applications were analysed. The leakage losses are a very important factor for the power output of such wheels and must be quantified. The theory developed by the authors would also allow to estimate the effect of blade number on leakage losses and thereby allow for a rational decision regarding this issue. The fact that there is no curved bed is also interesting since this is probably a more realistic scenario for application.

There are however several questions which need to be answered, and some additional work which would in the reviewer’s opinion greatly enhance the value of the work.

  1. Model tests:

The reviewer has two main questions regarding the model tests

2.1  The effect of ventilation?

Ventilation, i.e. the provision for air entering cells when water has to flow out is an important issue in waterwheel design.

The wheel in Fig.’s 3, 4 and 5 looks as if

  1. The blades are continuous from the outer rim to the central shaft, without a gap in between and
  2. There are side disks very close to the blades, or attached to the blades.

This is not clarified in the text. However, if (a) and (b) are correct, then the water cannot exit immediately into the downstream since air cannot get into the cells. The water can only start to leave when the tip of the blade exits the downstream water surface. This means that water will be lifted up, which would reduce the power output. This needs to be clarified.

2.2  Leakage calculations

Leakage is given as comparison measured and calculated, Fig. 6. The agreement of leakage model with measurements is good.

The leakage however needs to be quantified e.g. as %age of flow or power out to assess the difference to the efficiencies given in [14]. The reviewer’s rough estimates indicated that leakage accounts for a maximum of around 40% of the total flow, this factor therefore needs to be quantified and included.

The validated theoretical model for leakage can then be used to assess the effect of blade number on power out and efficiency. This allows to determine a realistic / improved efficiency for other blade numbers and an analysis of cost-effectiveness (blade nr vs efficiency increase). This needs to be included.

  1. Theory:

The theoretical efficiency curve in Fig, 10a gives a zero efficiency for a speed n→ 0. This appears not logical, since with the existence of significant leakage losses the efficiency should be negative for n = 0. Eq. 11-13 also leave the question of the efficiency for the condition of zero gap losses and speed n→ 0. Since the drag losses increase with n3, they can be expected to be very small for small values of n. The efficiency then becomes the ratio of Qw / Q where Qw ≈ Q, so that eta should approach 1.

The authors should clarify the efficiency calculations and ensure that leakage is included.

Author Response

We thank the reviewers for their suggestions and appreciations, which contributed to improve

the quality of the manuscript. Every comment is addressed following the reviewer’s comment (italic form). An extensive revision of English has been done.

Reviewer 1:

The manuscript describes an analysis of the Hydrostatic Pressure Wheel as a hydropower converter for very low head differences. The HPW is analysed theoretically, including leakage and turbulent losses. Laboratory experiments of an HPW are described and the results analysed. The model had 8 blades, and no curved section at the bed so that leakage losses can be expected to be high. The results are then used to assess the hydropower potential in an irrigation canal.

 General comments:

The paper is well written and illustrated. There are some minor linguistic errors, so the authors are advised to check the manuscript with a native English speaker.

There are some corrections in the annotated manuscript.

The paper’s contribution is significant, giving a much more thorough theoretical background than provided in the previous literature and addressing several design relevant aspects such as drag losses and leakage which were not considered before. Also, simplified geometries which will probably occur in real applications were analysed. The leakage losses are a very important factor for the power output of such wheels and must be quantified. The theory developed by the authors would also allow to estimate the effect of blade number on leakage losses and thereby allow for a rational decision regarding this issue. The fact that there is no curved bed is also interesting since this is probably a more realistic scenario for application.

There are however several questions which need to be answered, and some additional work which would in the reviewer’s opinion greatly enhance the value of the work.

  1. Model tests:

The reviewer has two main questions regarding the model tests 

2.1  The effect of ventilation?

Ventilation, i.e. the provision for air entering cells when water has to flow out is an important issue in waterwheel design.

The wheel in Fig.’s 3, 4 and 5 looks as if:

  1. The blades are continuous from the outer rim to the central shaft, without a gap in between and
  2. There are side disks very close to the blades, or attached to the blades.

This is not clarified in the text. However, if (a) and (b) are correct, then the water cannot exit immediately into the downstream since air cannot get into the cells. The water can only start to leave when the tip of the blade exits the downstream water surface. This means that water will be lifted up, which would reduce the power output. This needs to be clarified.

 Contrary to figure 1, there is no central axis and the air can go out by the top of the blade. In fact the blades are fixed by the side on the disks.

As a result, water may exit the blade even if the blade is still under water. However, some of the water is lifted up which decreases the efficiency. It is assumed that this phenomenon depends on the speed of rotation and is therefore included in the modeling of losses.

We added : » The blades are attached to the side disks. They are slightly shorter than the radius which allows the air and water pressures to be balanced between the blades”.

2.2  Leakage calculations

Leakage is given as comparison measured and calculated, Fig. 6. The agreement of leakage model with measurements is good.

The leakage however needs to be quantified e.g. as %age of flow or power out to assess the difference to the efficiencies given in [14]. The reviewer’s rough estimates indicated that leakage accounts for a maximum of around 40% of the total flow, this factor therefore needs to be quantified and included.

The percentage of lost flow is given by the value of RQ between 0 and 50%. We added : RQ is then the ratio of leakage. 

The validated theoretical model for leakage can then be used to assess the effect of blade number on power out and efficiency. This allows to determine a realistic / improved efficiency for other blade numbers and an analysis of cost-effectiveness (blade nr vs efficiency increase). This needs to be included.

Increasing the number of blades should reduce the w value. In the C1 configuration w should tend towards dl but it seems difficult to have a direct relationship between w and  the blade number. The fact that dl and w are already close means that 8 blades can already reduce the flow losses considerably. But it is not the case for C2 configuration.

  1. Theory:

The theoretical efficiency curve in Fig, 10a gives a zero efficiency for a speed n→ 0. This appears not logical, since with the existence of significant leakage losses the efficiency should be negative for n = 0. Eq. 11-13 also leave the question of the efficiency for the condition of zero gap losses and speed n→ 0. Since the drag losses increase with n3, they can be expected to be very small for small values of n. The efficiency then becomes the ratio of Qw / Q where Qw ≈ Q, so that eta should approach 1.

The authors should clarify the efficiency calculations and ensure that leakage is included.

When n tends to 0, the wheel does not rotate which makes the mechanical power tend to 0. In this case the ratio Qw/Q tends towards 0 and not towards 1 because most of the flow passes through the leaks. Even if this is not taken into account in the model, we can also assume that the leakage (w) is even greater when the wheel rotates too slowly. We have added in the text :  “when n tends to  0, the efficiency also tends to 0 because the leakage flow rate becomes equal to the total flow rate (RQ ~ 0). Indeed, the wheel behaves like a sluice gate and no mechanical power is obtained.”

Reviewer 2 Report

The manuscript is well organized and the finding and limitations of the work are adequately described. I just have a few minor comments:

  • The authors may wish to add a figure showing different types of turbines, water wheels to aid the discussion presented on page 2.
  • Rests are language editing and formatting comments:
    • Line 37: 'Regarding [3]', the reference should be placed at the end of the line rather than here?
    • Line 82: Should be 'Special attention', rather than ' A special attention'.
    • While citing figures, tables, the authors used 'cf.'. As far as I know, cf. means compare. Hence, I think it is not necessary to use this term while citing the figures and tables.
    • First-line in 2.1 'The Hydrostatic Pressure Wheel developed by [14] consists in a wheel made up of radial blades which are longer than the inlet water height (cf. figure 1).' can be rewritten as 'As shown in Figure 1, the HPW developed in [14] consists of a wheel made up of radial blades longer than the inlet water height.'
    • Line 106: should be ' ... and the flow conditions'. Similarly in line 172.
    • The adverbs are not used properly on several occasions: For example in Line 77 - should be 'to control water levels accurately', line 101 - should be 'to use the wheel efficiently', line 116 - should be 'the theory is rewritten here', etc. The entire manuscript should be revised carefully to rectify these issues.
    • Line 135: '... the gap dl is note directly used. Indeed, an equivalent opening...', should be 'the gap dl is not directly used. Instead, an equivalent opening ...'
    • Section 3 heading can be renamed as 'Experimental setup'?
    • In the paragraph below Figure 3: the term 'break' should be replaced with 'brake' everywhere.
    • Please check the sentence in lines 218 to 219.
    • LIne 239: should be 'can be seen' instead of 'can be shown'. Similar issues exist in other places.

Please revise the manuscript to improve the English.

Author Response

We thank the reviewers for their suggestions and appreciations, which contributed to improve the quality of the manuscript. Every comment is addressed following the reviewer’s comment (italic form). An extensive revision of English has been done.

Reviewer 2 :

The manuscript is well organized and the finding and limitations of the work are adequately described. I just have a few minor comments:

  • The authors may wish to add a figure showing different types of turbines, water wheels to aid the discussion presented on page 2.

  • Rests are language editing and formatting comments:
    • Line 37: 'Regarding [3]', the reference should be placed at the end of the line rather than here?

Modification done

    • Line 82: Should be 'Special attention', rather than ' A special attention'.

Modification done

    • While citing figures, tables, the authors used 'cf.'. As far as I know, cf. means compare. Hence, I think it is not necessary to use this term while citing the figures and tables.

Modification done

    • First-line in 2.1 'The Hydrostatic Pressure Wheel developed by [14] consists in a wheel made up of radial blades which are longer than the inlet water height (cf. figure 1).' can be rewritten as 'As shown in Figure 1, the HPW developed in [14] consists of a wheel made up of radial blades longer than the inlet water height.'

Modification done

    • Line 106: should be ' ... and the flow conditions'. Similarly in line 172.

Modification done

    • The adverbs are not used properly on several occasions: For example in Line 77 - should be 'to control water levels accurately', line 101 - should be 'to use the wheel efficiently', line 116 - should be 'the theory is rewritten here', etc. The entire manuscript should be revised carefully to rectify these issues.

Modification done

    • Line 135: '... the gap dl is note directly used. Indeed, an equivalent opening...', should be 'the gap dl is not directly used. Instead, an equivalent opening ...'

Modification done

    • Section 3 heading can be renamed as 'Experimental setup'?

Modification done

    • In the paragraph below Figure 3: the term 'break' should be replaced with 'brake' everywhere.

Modification done

    • Please check the sentence in lines 218 to 219.

The sentence has been changed :”It appears that this torque increases linearly with speed and its value remains close to zero..”

    • LIne 239: should be 'can be seen' instead of 'can be shown'. Similar issues exist in other places.

Modification done
